# Neuronal processes and glial precursors form a scaffold for wiring the developing mouse cochlea

N. R. Druckenbrod[1,2], E. B. Hale[1], O. O. Olukoya [1], W. E. Shatzer[1] & L. V. Goodrich [1✉]

In the developing nervous system, axons navigate through complex terrains that change depending on when and where outgrowth begins. For instance, in the developing cochlea, spiral ganglion neurons extend their peripheral processes through a growing and heterogeneous environment en route to their final targets, the hair cells. Although the basic principles of axon guidance are well established, it remains unclear how axons adjust strategies over time and space. Here, we show that neurons with different positions in the spiral ganglion employ different guidance mechanisms, with evidence for both glia-guided growth and fasciculation along a neuronal scaffold. Processes from neurons in the rear of the ganglion are more directed and grow faster than those from neurons at the border of the ganglion. Further, processes at the wavefront grow more efficiently when in contact with glial precursors growing ahead of them. These findings suggest a tiered mechanism for reliable axon guidance.

[1] Department of Neurobiology, Harvard Medical School, 220 Longwood Avenue, Boston, MA 02115, USA. [2] Present address: Decibel Therapeutics, 1325 Boylston St #500, Boston, MA 02215, USA. ✉email: Lisa_Goodrich@hms.harvard.edu

Neuroscientists have long puzzled how neurons make the connections needed for proper circuit function, with mechanical explanations by Weiss[1] set aside in favor of Sperry's chemoaffinity hypothesis[2]. With the discovery of axon guidance molecules, the field coalesced around the idea that axons are guided by a combination of attractive and repulsive cues that act at short or long range, with direction specified by target-derived gradients[3]. However, mathematical modeling studies suggest that chemoattractive gradients are not solely responsible for the remarkable precision of guidance events in vivo, where axons grow through complex and changing environments[4]. Although synergy among cues may improve fidelity, other cellular mechanisms likely contribute, such as fasciculation with pioneers and avoidance of repellant boundaries[5]. For instance, in the spinal cord, commissural axons grow along a permissive substrate of Netrin-1 in the subpial region before turning toward an instructive gradient of Netrin-1 and other cues emanating from the floor plate[6–9]. Thus, axons may rely on different mechanisms as they move through distinct terrains.

The cochlea presents a distinct landscape for axon growth compared to the spinal cord, with neural processes organized into a spatial stereotyped pattern within a highly heterogeneous cellular environment. The cochlea is comprised of three fluid-filled ducts: scala vestibuli, scala media, and scala tympani. The auditory sensory epithelium, the organ of Corti, sits on the floor of scala media and vibrates in response to wavelengths of sound, thereby activating sensory hair cells. Information is transmitted to the central nervous system by the spiral ganglion neurons (SGNs). SGNs are bipolar neurons that extend a peripheral process to innervate hair cells in the organ of Corti and a central process that innervates target neurons in the auditory brainstem. In the cochlea, hair cells and SGNs are arranged tonotopically, from high frequencies in the base to low frequencies in the apex. Hence, tonotopy is represented in the spatial arrangement of the SGN peripheral processes, which are arrayed in bundles that radiate out like spokes of a wheel. The radial bundles of SGN processes are separated from each other by the mesenchymal cells of the osseous spiral lamina (OSL). Glia are also prominent in the cochlea, both throughout the ganglion, where satellite cells myelinate SGN cell bodies in mice[10], and within the radial bundles, where Schwann cells ensheath processes from Type I SGNs. Type I SGNs innervate the inner hair cells (IHCs) and comprise ~95% of the population[11]; these neurons are primarily responsible for encoding sound. The remaining Type II SGNs extend unmyelinated processes that spiral among the outer hair cells (OHCs) where they are proposed to play a role in sensing damage[12]. There are also glial non-sensory cells surrounding the hair cells in the organ of Corti. Like the Schwann cells and satellite glia, these supporting cells express PLP[13,14]. However, they are derived from the otic placode[15], whereas Schwann cells and satellite cells develop from the neural crest.

Target-derived cues seem to play a minimal role in gross cochlear wiring[16,17]. For instance, in mice with no differentiated hair cells, such as *Atoh1* mutants, SGN peripheral processes still form radial bundles and reach the organ of Corti[18,19]. In addition, although the classic chemoattractant Netrin-1 can elicit outgrowth in vitro[20], it is not expressed in the organ of Corti and is not required for cochlear wiring in vivo[21]. On the other hand, there is a prominent role for repulsion[22–25]. In the OSL, Ephrin-Eph signaling between the neurons and the mesenchyme ensures fasciculation of SGN processes in radial bundles[24], whereas in the organ of Corti, Semaphorin 3F and Ephrin A5 keep the processes from entering the OHC region[23,25]. Thus, once the processes start growing, they may be kept on course by repulsive cues encountered along the way. How peripheral axons begin this journey, however, remains unknown.

Several observations suggest that glia might be involved in the earliest stages of cochlear wiring. By E10.5 in mouse, neural-crest-derived glial precursors have migrated from the pharyngeal arches into the otic vesicle, where they will differentiate into Schwann cells and satellite glia[26]. Classic histological studies showed that SGN peripheral processes pass through a glial funnel[27]; the identity of these cells was not determined. Likewise, three-dimensional reconstructions revealed interdigitation of glial precursors with the earliest growing neurites in the otic vesicle[26]. There are also hints that glial precursors are necessary for normal cochlear innervation. For example, in mice with impaired invasion of the cochlea by neural-crest cell-derived glial precursors, radial bundles do not form normally[13,28]. However, in these animals, the SGNs are also mispositioned, making it hard to know whether the abnormal radial outgrowth is secondary to an earlier migration phenotype.

Here, we define the cellular mechanisms underlying the earliest stages of SGN peripheral process growth, including a potential role for neural-crest-derived glial precursors. Using a combination of time-lapse imaging and three-dimensional reconstructions of individual SGN processes in the context of the intact developing cochlea, we show that SGN neurites interact with each other and with glial precursors during radial bundle formation. The timing and nature of these interactions suggest that neuron–glia interactions establish a scaffold for subsequent neuron–neuron interactions. These findings highlight the multi-stepped nature of axon guidance in vivo and can inform efforts to re-wire the damaged cochlea.

## Results

**Early SGN peripheral processes extend along glial precursors.** Cochlear wiring occurs in a complex and dynamic environment, with Type I and Type II SGN processes growing toward the organ of Corti even as the cochlea lengthens and coils[29]. SGNs originate in the otic vesicle between E9 and E12 in mouse and delaminate into the surrounding mesenchyme[30,31]. Differentiating SGNs extend peripheral processes back toward the organ of Corti, reaching nascent hair cells around E15[32] and forming synapses by birth[33]. There is a topographic gradient of development, with SGNs in the base maturing before those in the apex. Neural-crest-derived precursors for Schwann cells and satellite glia are present from the earliest stages of neurite outgrowth[26]. Thus, long before reaching target hair cells in the organ of Corti, the SGN peripheral processes interact with other neurons and their processes, as well as with glial precursors and mesenchyme.

To gain insights into the cellular mechanisms that might influence the growth of SGN peripheral processes toward the cochlear duct, we characterized the wavefront of peripheral process growth relative to glial precursors, using anti-β-III tubulin (TuJ) or anti-Neurofilament antibodies to label the SGN processes and a *PLP-GFP* transgenic reporter[34] to label glial precursors. We focused on E14-E14.5, which is when most processes cross the border of the ganglion[24], with radial bundles apparent around E15-E15.5[32]. At these stages, PLP-GFP marks cells in the ganglion and OSL that match the features of neural-crest-derived glial precursors;[26] PLP-GFP + glial supporting cells are confined to the developing sensory epithelium, far away from the region of interest (Fig. 1c, h). Thus, the PLP-GFP+ cells imaged throughout this study likely correspond to precursors for satellite cells and Schwann cells.

We found that early SGN processes interact with neural-crest-derived glial precursors as they begin to grow through the OSL. In addition to the expected intermingling of glial precursors and SGN cell bodies within the ganglion, the SGN peripheral processes are closely associated with a distinct population of

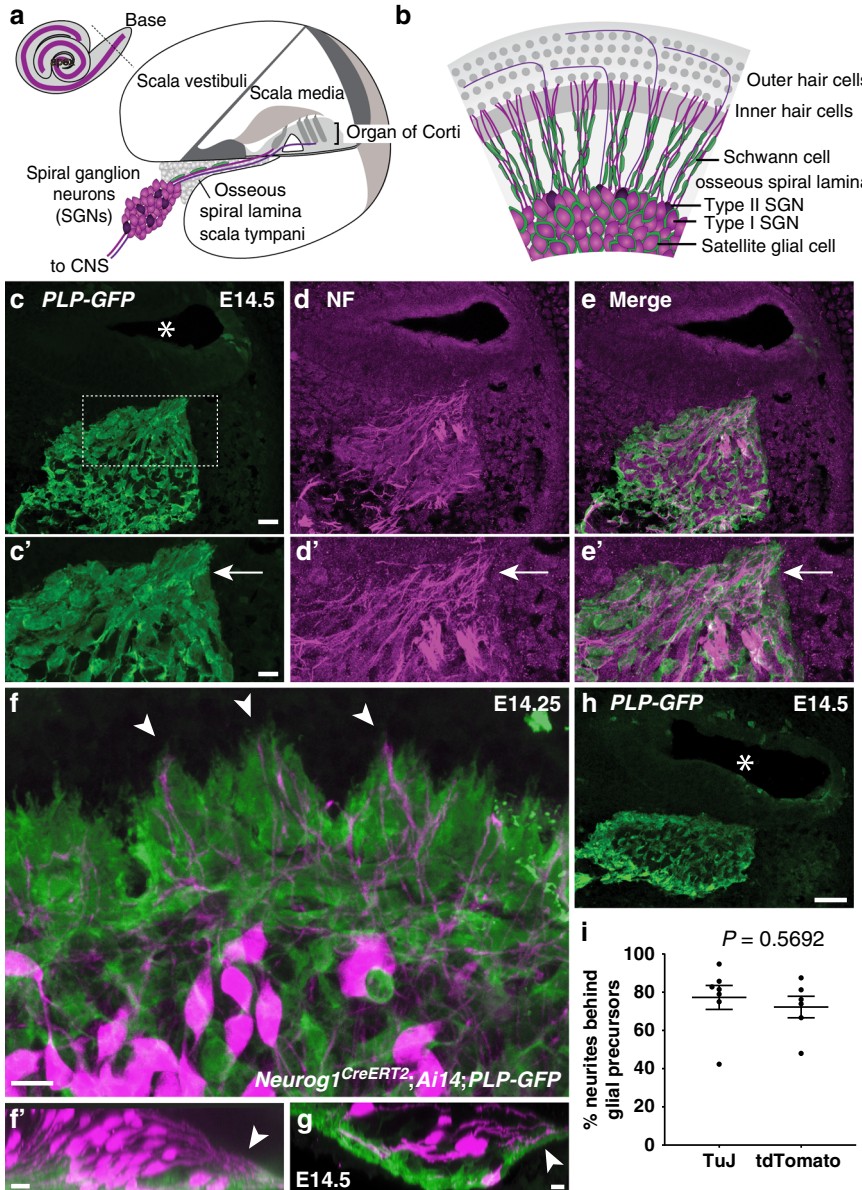

**Fig. 1 Distal SGN peripheral processes are closely associated with glia. a, b** As shown in transverse (**a**) and top-down diagrams, Type I (magenta) and Type II (purple) spiral ganglion neurons (SGNs) extend peripheral processes through the osseous spiral lamina (gray) to the organ of Corti, which is housed in scala media, one of three fluid-filled chambers in the cochlea. Their central axons project to the central nervous system (CNS). Type I SGN cell bodies and peripheral processes (magenta) are myelinated by satellite glial cells and Schwann cells (green). **c–e** Transverse section through an E14.5 PLP-GFP animal stained for GFP (green, **c**) and neurofilament (NF) (magenta, **d**; merged, **e**), with magnified views of the boxed region (**c**) shown in (**c'–e'**). PLP-GFP + glial precursors form a path to the cochlea for NF + neurites (arrows, **c'–e'**). Nine cochleae from 6 animals were examined. **f, g** Wholemount E14.25 *Neurog1^{CreERT2};Ai14;PLP-GFP* cochlea stained for tdTomato (magenta) and GFP (green), shown top-down (**f**; organ of Corti is up and the ganglion is down) and from the side (**f'**; organ of Corti is right and the ganglion is left). Arrowheads (**f**) indicate three tdTomato-labeled peripheral processes that are preceded by glial processes at the wavefront of growth. This close relationship between the distalmost SGN peripheral processes (arrowhead, **f'**) and PLP-GFP+ glial precursors is highlighted in a side view (**f'**) and in a single confocal slice from a stack through an E14.5 *Neurog1^{CreERT2};Ai14;PLP-GFP* wholemount cochlea (arrowhead, **g**). 6 cochleae were imaged. **h** Intensely GFP+ glial precursors surround the ganglion, as seen in a cross-section through an E14.5 *PLP-GFP* head stained for GFP (green). **i** SGN peripheral processes were scored for whether they were ahead of or behind a glial precursor in E14-E14.5 *PLP-GFP* cochleae stained for TuJ and GFP (N = 7 cochleae; 164 neurites) or E14-E14.5 *Neurog1^{CreERT2};Ai14;PLP-GFP* cochleae stained for tdTomato and GFP (N = 6 cochleae; 125 neurites). The percent of neurites behind glial precursors is plotted for each cochlea, overlaid with the mean ± standard error of the mean (SEM). Regardless of how neurites were labeled, the majority grow behind a process from a glial precursor, with no statistical difference between percentages obtained using each method (P = 0.5692, unpaired, two-tailed t-test). Source data are provided as a Source data file. Asterisk (\*), lumen of cochlear duct (**c, h**). Scale bars: 40 µm (**a–c, f**), 10 µm (**a'–c', d–e**).

glial precursors that protrudes from the ganglion and forms a bridge to the cochlear duct (Fig. 1a, c–e). This population stands out both for its morphology, with large flat cells, and for its more intense PLP-GFP signal. At this stage, it is not yet possible to tell which of the PLP-GFP + progenitors will differentiate as Schwann cells or satellite cells.

To further define the relationship between the extending peripheral processes and glial precursors, we imaged E14.25-E14.5 cochleae from *Neurog1*[CreERT2]*;Ai14;PLP-GFP* animals, in which a random subset of SGNs express tdTomato and all glial precursors are GFP+. At this early stage of development, no peripheral processes have grown more than 50 μm past the border of the ganglion. Wholemount preparations confirmed that the SGN peripheral processes extend along intensely PLP-GFP-positive glial precursors (Fig. 1b, f, g). Indeed, a population of strongly stained glial precursors seems to form a shell around the entire ganglion, though it is possible that the intensity of the signal is simply higher because they are not interspersed with neuronal cell bodies here (Fig. 1h). Quantification showed that most peripheral processes are preceded by glial precursors: 72.25% ± 5.63 (s.e.m.) of the most distal tdTomato-labeled SGN peripheral processes in the E14 cochlea were preceded by a PLP-GFP positive glial precursor ($n = 125$ neurites from 6 cochleae) (Fig. 1i). A similar relationship was observed when β-III tubulin+ peripheral processes were analyzed (77.28% ± 6.27 s.e.m.) ($n = 164$ neurites from 7 cochleae). Since similar percentages were observed when quantifying all β-III tubulin+ processes or only those randomly labeled by *Neurog1*[CreERT2] recombination of *Ai14*, the behavior of tdTomato+ processes seems to be typical. In addition, since both methods yielded similar results, it is likely that we visualized the entire SGN terminal, though we cannot rule out the presence of very fine processes that are not filled with either tdTomato or β-III tubulin. These observations extend previous reports that neural-crest-derived glial precursors are closely affiliated with SGN and VGN processes during early stages of inner ear innervation[26], as well as the classic observation of a glial funnel through which early SGN processes appear to be directed toward the otic vesicle[27]. Thus, glial precursors are poised to influence the initial outgrowth of SGN peripheral processes.

**Early SGN processes exhibit diverse outgrowth behaviors.** Although the earliest SGN peripheral processes seem closely aligned with glial precursors in the funnel, other processes were positioned further away, raising the question of whether all SGNs follow the same path. To learn more about the range of trajectories taken by different SGNs, we made three-dimensional models of 151 SGNs from the mid-base of the cochlea in E14-E14.5 *Neurog1*[CreERT2]*;Ai14* animals ($n = 7$) (Fig. 2a, b; Supp. Fig. 1 and Supp. Video 1). Consistent with the impression that SGNs grow differently depending on their local environment, we observed a range of morphologies that correlated with the position of each neuron's cell body within the ganglion. SGNs whose cell bodies were close to the border extended twisted, short neurites (orange, Fig. 2b′). By contrast, those whose cell bodies were in the rear of the ganglion exhibited long and directed neurites (purple, Fig. 2b′). SGNs in the middle of the ganglion showed intermediate morphologies (green, Fig. 2b′). In addition, rear SGNs extended peripheral processes along a flat trajectory and were at the bottom of the pile of processes (Fig. 2b″), closest to the intense PLP-GFP+ cells that form the funnel. By contrast, processes from the SGNs at the border were positioned at a steeper slope, diving down to meet the processes from the middle and rear SGNs. These qualitative observations were confirmed by quantitative analysis of SGNs whose cell bodies sat near the border (border cells, $n = 28$), in the middle (middle cells, $n = 52$), or in the rear (rear cells, $n = 71$) (Fig. 2c–e and Supp. Fig. 2), using the peripheral circumferential border of the ganglion as a reference (Fig. 2a). The border cells showed consistently less directed, shorter processes that were at a steeper slope than those from the rear cells, with the middle cells falling in between (see Fig. 2 legend for means and SEM). Thus, SGN peripheral processes follow distinct paths depending on the position of their cell body in the ganglion: the rear SGN processes are closest to the glial precursors, with middle and border SGN processes layering on top.

As predicted from analysis of fixed tissue, time-lapse imaging in embryonic cochlear explants revealed that SGN peripheral processes change their outgrowth behavior as they progress from the ganglion toward the developing organ of Corti. Since this entire journey takes place from E14 to E15.5 in vivo, we imaged the overall wavefront of outgrowth in *Bhlhb5*[cre]*;Ai14* cochleae at three separate positions along the trajectory: as the peripheral processes first exit the ganglion (at E14.25), as they are pushing through the OSL (around E15) and upon reaching the organ of Corti (at E15.5). Because the cochlear duct is still lengthening throughout this time, groups of SGNs occasionally moved, consistent with the appearance of a few longitudinal cell bodies in fixed tissue (Fig. 2b). However, we did not observe any translocation of individual SGN somas that would indicate active migration along the length of the cochlea. At each stage, we analyzed outgrowth across a 50 μm region at the wavefront: immediately adjacent to the ganglion (R1) (Supp. Movie 2), within the developing OSL (R2) (Supp. Movie 3), and near the nascent organ of Corti (R3) (Supp. Movie 4) (Fig. 3a). The position of the tip of each process was plotted every 7–14 min (Fig. 3b, c), such that we could calculate the speed and direction of movement from the origin to the final position ($n = 3$ cochleae) (Fig. 3d, e). Speed was calculated by dividing the distance between the origin and the final position over time. Directionality was quantified as the length of the entire path divided by the length from origin to final position. We found that the speed and directionality of SGN process outgrowth varied along the trajectory. In R1, SGN peripheral processes ($n = 55$) made slow progress (Fig. 3d) (speed: 0.205 ± 0.009 μm/min, mean ± SEM) and followed tortuous paths with many changes in direction (Fig. 3e) (directionality index: 0.476 ± 0.032). By contrast, SGN process outgrowth was faster (0.337 ± 0.011 μm/min) and had a higher directionality index (0.875 ± 0.017) within R2 ($n = 57$), before slowing down (0.190 ± 0.011 μm/min) and becoming less directed (0.468 ± 0.031) in R3 ($n = 45$). The behavior in R3 was similar to the exploratory behavior we observed previously in analysis of SGN peripheral processes close to the organ of Corti at E16.5[22]. Thus, even though the peripheral processes follow a relatively straight path toward the organ of Corti, they do not grow in the same way at each point along the trajectory.

Closer examination of R1 revealed that even within a single location, the earliest extending SGN peripheral processes showed heterogeneous behaviors (Fig. 3f). While many processes showed slow, exploratory behavior consistent with the average behavior of the group, others grew in a fast, directed manner, occasionally darting ahead of the wavefront of neurite outgrowth (Fig. 3g and Supp. Movie 2). Indeed, a subset of processes (14 neurites of 55 total, 25.5%) showed directionality similar to that in R2 (Fig. 3c, e, f). The remaining neurites followed more convoluted paths as they left the ganglion (Fig. 3f). Since SGN process morphology and trajectory varies with position (Fig. 2), these observations suggest that processes respond to local cues in their environment that affect how they exit the ganglion and grow through the mesenchyme.

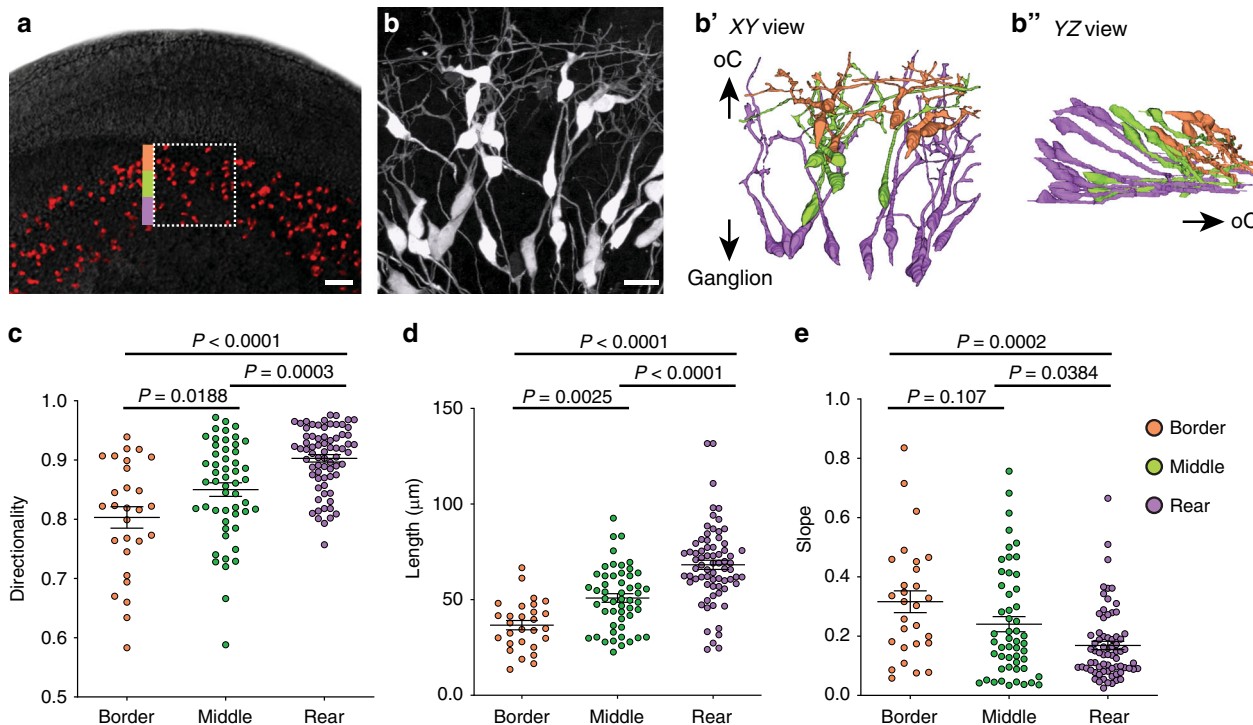

**Fig. 2 SGN morphology varies with cell body position. a** A top-down view of a wholemount cochlea from an E14.5 *Neurog1^CreERT2^;Ai14* animal, where a random subset of SGNs express tdTomato (red). SGN morphology was analyzed at three positions in the ganglion: close to the border (orange), in the middle (green), or in the rear (purple). **b** A high-power view of the boxed area in (**a**), with reconstructions of individual SGNs shown in (**b'**) (XY view) and (**b"**) (YZ view). SGNs are color-coded according to the position of their cell body. 7 cochleae were stained and analyzed (see Supp. Figs. 1 and 2). **c–e** Quantification of SGN morphology. Border cells ($n = 28$) (orange dots) extend processes that are less directional (**c**), shorter (**d**), and have a greater slope (**e**) than rear cells ($n = 71$). Middle cells ($n = 52$) exhibit intermediate morphologies. Plots show raw data with means ± SEM superimposed. Directionality: border cells: 0.803 ± 0.012; middle cells: 0.850 ± 0.011; rear cells: 0.903 ± 0.006. Length: border cells: 36.68 ± 2.433; middle cells: 50.88 ± 2.247; rear cells: 68.22 ± 2.418. Slope: border cells: 0.316 ± 0.037; middle cells: 0.240 ± 0.026; rear cells: 0.168 ± 0.014. Significance was assessed using ANOVA with Tukey's multiple comparison test. Directionality: border vs middle, $P = 0.188$; middle vs rear, $P = 0.0003$; border vs rear, $P < 0.0001$. Length: border vs middle, $P = 0.0025$; middle vs rear, $P < 0.0001$; border vs rear, $P < 0.0001$. Slope: border vs middle, $P = 0.107$; middle vs rear, $P = 0.0384$; border vs rear, $P = 0.0002$. Source data are provided as a Source data file. Scale bars: 50 μm (**a**) and 25 μm (**b**). See Supplementary Movie 1.

**SGN processes grow differently depending on their position.** Since analysis of morphologies revealed systematic differences among rear, border, and middle SGNs, we hypothesized that SGN peripheral processes grow differently depending on their location in the ganglion and thus what kind of environment they first encounter. To characterize the behavior of individual axons extending from the rear or border of the ganglion, we performed time-lapse imaging of E14-E14.5 cochleae from *Neurog1^CreERT2^; Ai14* animals. Because SGNs are sparsely labeled in this strain, we were able to define the cell body position for each extending peripheral process in R1. Consistent with the position-dependent variation in SGN morphology, border and rear SGNs showed strikingly different patterns of peripheral process outgrowth (Supp. Movie 5). The peripheral processes from the rear SGNs followed directed paths (blue arrows, Fig. 4a). By contrast, border SGNs instead showed exploratory, undirected outgrowth (orange dots, Fig. 4a). Quantification confirmed that trajectories from rear cells were significantly more directional than those from border cells (directionality index [mean ± SEM]: 0.710 ± 0.036 for rear cells vs 0.401 ± 0.035 for border cells) ($P < 0.0001$ by unpaired, two-tailed *t*-test; $n = 25$ border cells and 24 rear cells from $N = 3$ cochleae).

Analysis of outgrowth behavior in R2 revealed additional heterogeneity here, as well. In this case, the processes had grown too far away from the ganglion to link them definitively to border or rear SGN cell bodies. Instead, we compared the behavior of processes at the wavefront to those approaching from behind

(Fig. 4c and Supp. Movie 6). We found that in this region, the processes at the wavefront (orange arrowheads, Fig. 4c) were significantly less directed than the processes behind them (blue arrowheads, Fig. 4c) (Fig. 4d) (directionality index [mean ± SEM]: 0.456 ± 0.025 at wavefront vs 0.825 ± 0.156 behind the wavefront) ($P < 0.0001$ by unpaired, two-tailed *t*-test; $n = 17$ wavefront neurites and 23 follower neurites from $N = 3$ cochleae). At this point in the trajectory, it seems that the trailing processes have fasciculated with the leaders, which are still navigating forward through the mesenchyme. It was not possible to quantify the degree of fasciculation, as only a subset of processes is labeled. However, it is already established that fasciculation occurs in this region of the mesenchyme[24]. Thus, within both R1 and R2, SGN processes use different cellular mechanisms to approach the same targets, with some processes seeming to interact with non-neuronal cells in the environment and others relying on fasciculation with other SGN processes that have already forged the way.

**SGN processes interact with migrating glial precursors.** Although axon-axon fasciculation is a well-established mechanism for nervous system wiring, the potential influence of neuron–glia interactions is not well understood. Several observations suggest that glial precursors may encourage innervation by the earliest processes and hence create a scaffold for later arriving processes. First, PLP-GFP + cells precede most neurites at the wavefront (Fig. 1f, i). Second, the processes from the rear

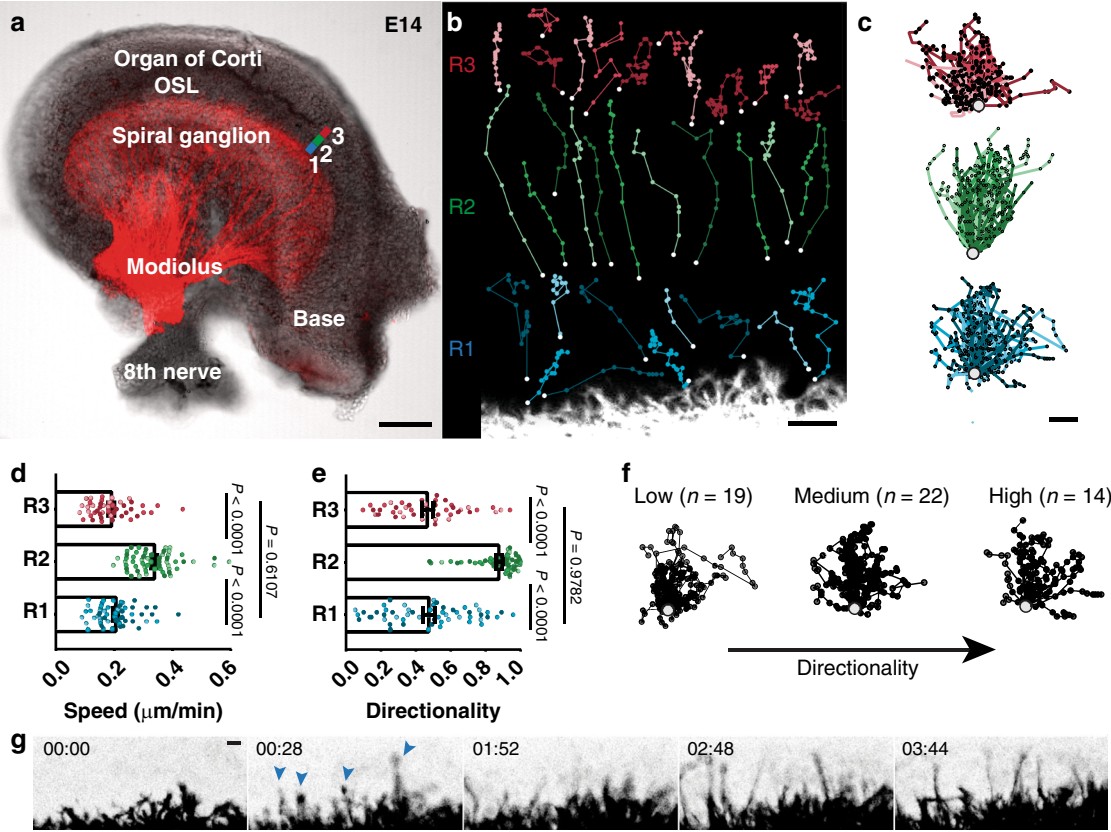

**Fig. 3 SGN peripheral processes exhibit a variety of outgrowth behaviors. a** A wholemount E14 cochlea, with SGNs genetically labeled red using *Bhlhb5*[cre] and an *Ai14* tdTomato reporter. Process outgrowth was imaged in three different regions of E14–E15 cochleae. Region 1 (blue, R1) is immediately adjacent to the ganglion; region 2 (green, R2) is in the developing osseous spiral lamina (OSL), and region 3 (red, R3) is close to the organ of Corti. See Supplementary Movies 2–4. **b** Representative trajectories of processes imaged as they grew through each region. **c** All trajectories were collapsed onto a common origin, revealing overall differences in the pattern of growth in each region. $N = 3$ cochleae for each region with $n = 55$ R1 trajectories, 64 R2 trajectories, and 44 R3 trajectories. **d**, **e** Quantification of the speed (**d**) and directionality (**e**) of SGN peripheral process outgrowth in R1, R2, and R3 (shown as raw data with mean ± SEM indicated). Growth is faster and more directed in R2 than in R1 or R3. Within R1, some SGN processes are as directed as those in R2, with directionality indices greater than 0.66. Significance was assessed using ANOVA with Tukey's multiple comparison test. R1 vs R2 speed, $P < 0.0001$; R1 vs R2 directionality, $P < 0.0001$; R1 vs R3 directionality, $P = 0.9782$; R1 vs R3 speed, $P = 0.6107$; R2 vs R3 directionality, $P < 0.0001$; and R2 vs R3 speed, $P < 0.0001$. **f** R1 trajectories were grouped according to directionality, from low (0–0.33) to medium (0.33–0.66) to high (0.66–1). The number of trajectories in each group is indicated. Source data are provided as a Source data file. **g** Frames from a video (Supp. Movie 2) showing a subset of SGN peripheral processes (arrowheads) that grow ahead of the overall wavefront. Time is indicated in hh:mm. Scale bars: 100 μm (**a**), 10 μm (**b, c, g**).

SGNs are closely affiliated with the glial funnel (Fig. 1f′, g and Fig. 2b″) and also grow in a more directed manner (Fig. 4b). Third, when neural-crest-derived glia are depleted from the cochlea, radial bundle formation is severely disrupted[13,28].

To learn more about how interactions with glial precursors might affect axon growth, we simultaneously imaged SGN peripheral processes and glial precursors in E14–E14.5 cochleae from *Neurog1*[CreERT2];*Ai14;PLP-GFP* animals ($N = 4$), characterizing both the interactions at the wavefront and behind, where a scaffold is already in place. We found that the PLP-GFP + glial precursors migrate in radial chains through the mesenchyme toward the sensory epithelium (Fig. 5a), consistent with the idea that they correspond to neural-crest-derived glial precursors for Schwann cells. Within the chains, the SGN peripheral processes sometimes grew directly on the glial precursors, such that tdTomato and GFP signals co-localized. At other times, the tdTomato and GFP signals did not overlap. Qualitatively, the neurites and glial precursors seemed to grow in tandem in a dynamic fashion, with neurites occasionally observed without any nearby glial precursors and vice versa. This is consistent with results from fixed tissue, where ~20% of neurites extended beyond glial precursors at the wavefront (Fig. 1i).

To gain a better sense of how peripheral process growth might be influenced by interactions with the glial precursors, we quantified the speed and directionality of neurites while they were growing either on ($n = 18$ tracks from 4 cochleae) or off ($n = 17$ tracks from 4 cochleae) the glial precursors, as defined by co-localization of tdTomato and GFP signals. Quantification revealed that the neurites grew quickly and in a directed manner when in contact with glial precursors (Fig. 5b, c). By contrast, SGN processes that were not contacting glial precursors were slower and more exploratory, with a significantly lower direction index (speed [mean ± SEM]: $0.451 \pm 0.044$ μm/min when on glial precursors and $0.324 \pm 0.041$ when off glial precursors; directionality index [mean ± SEM]: $0.748 \pm 0.053$ when on glial precursors and $0.210 \pm 0.042$ when off glial precursors). Thus, SGN processes make a more efficient path toward the organ of Corti when they are interacting with glial precursors. Furthermore, there were several cases where SGN neurite growth was predicted by a similar change in glial precursor behavior, as if the SGN neurites are guided by the glial precursors (Supp. Movies 7–11). For example, we observed multiple examples where an SGN peripheral process retracted (*, Fig. 5d, g) shortly after the associated PLP-GFP positive glial process had retracted, such that

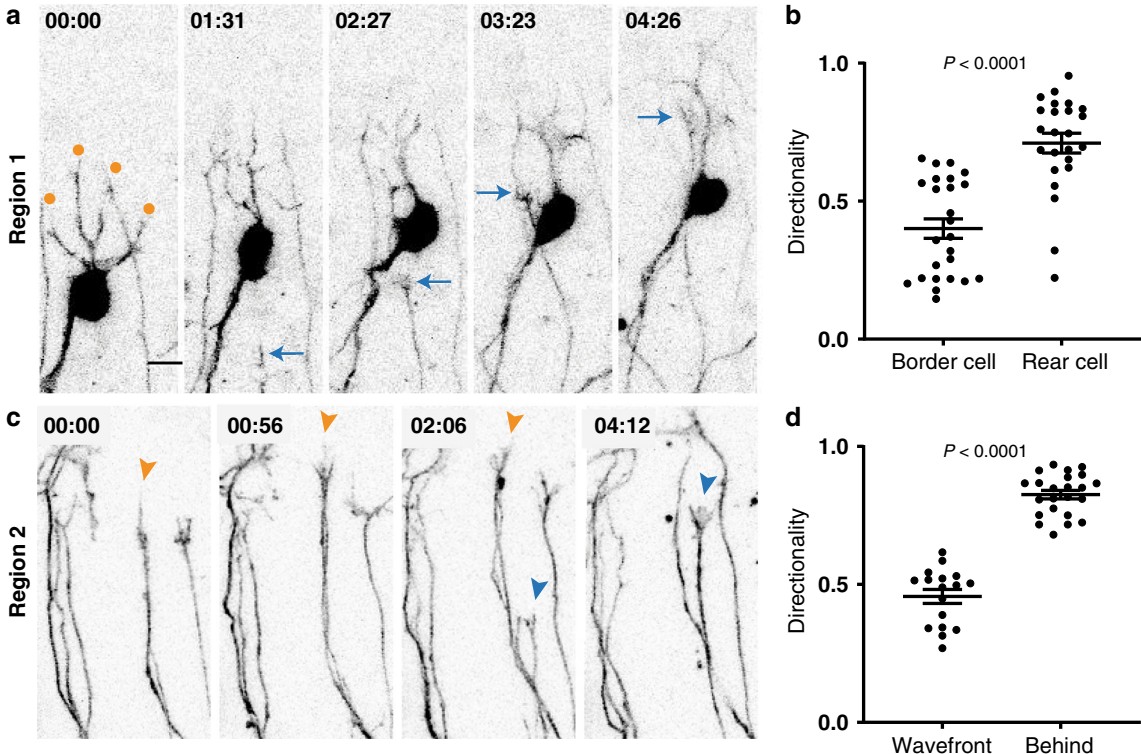

**Fig. 4 Directionality differs among SGN processes growing through similar regions. a–d** Frames from movies of sparsely labeled tdTomato+ SGNs from *Neurog1^CreERT2;Ai14* animals as they grow in Region 1 (**a**) (*N* = 3 cochleae) (Supp. Movie 5) or Region 2 (**c**) (*N* = 3 cochleae) (Supp. Movie 6), with directionality quantified in (**b**) and (**d**). In Region 1 (**a**), a border SGN extends highly branched processes (orange dots) whereas a peripheral process from a rear cell (cell body out of view) (blue arrows) follows a rapid and directed path. Quantification of 25 border and 24 rear cells (**b**) confirmed that peripheral processes from rear cells are significantly more directed. In Region 2 (**c**), an SGN process at the wavefront (orange arrowheads) extends many branches and makes little progress. An SGN process arriving from behind (blue arrowheads) moves rapidly and is capped by a large, unbranched growth cone. Quantification (**d**) revealed significantly more directed growth of processes behind the wavefront (*n* = 17 wavefront processes and 23 trailing processes). Note that not all SGN processes are labeled, so it is not possible to detect fasciculation events reliably. Raw data are plotted, with mean ± SEM superimposed. Significance was assessed using an unpaired, two-tailed *t*-test. Directionality: border vs rear, *P* < 0.0001; wavefront vs behind, *P* < 0.0001. Source data are provided as a Source data file. Time is indicated in hh:mm. Scale bar = 10 μm (**a**, **c**).

the SGN and glial processes ended up once again on top of each other. Likewise, forward movement of a PLP-GFP+ process was rapidly followed by forward movement of the associated SGN peripheral process in the same direction (+, Fig. 5e). In other cases, the SGN processes grow along pre-existing glial tracts (arrowheads, Fig. 5f, g). This may be especially true behind the wavefront of growth, where the glia seem less active and may instead provide a permissive substrate. Thus, SGN peripheral process outgrowth appears to be facilitated by interactions with glial precursors that are migrating in the same direction.

## Discussion

Although there is abundant evidence that axons can be directed to their targets by a combination of positive and negative cues, the full range of behaviors that ensure reliable axon guidance in vivo remains to be defined. We examined this question by characterizing outgrowth in the cochlea, where axons grow through a heterogeneous environment to reach their targets in the sensory epithelium. In this cellular landscape, processes exhibit different patterns of outgrowth depending on their position within the ganglion. For instance, during early cochlear wiring, SGNs in the rear of the ganglion extend processes that have easy access to a distinct population of glial precursors that bridge the ganglion to the cochlear duct. The processes grow faster and make fewer changes in direction when contacting the glial precursors in this region. Follower SGNs closer to the border of the ganglion, on the

other hand, must grow down toward this path, where they are able to fasciculate with the processes from the rear. Moreover, imaging studies raise the possibility that glial precursors actively direct axon outgrowth toward the organ of Corti, as well as providing an attractive, permissive surface. Together, our findings support a model in which glial precursors enable rapid and directed growth of SGN peripheral processes toward the organ of Corti. Such a mechanism provides a straightforward way to establish the basic topography of the cochlea. Rather than sensing cues that tell them where they are along the apical-basal axis, individual SGN peripheral processes may simply need to follow a straight path toward the organ of Corti, with growth encouraged by glial precursors and corralled by surrounding mesenchyme[24]. In addition, with the flexibility to grow either along glial precursors or along each other, SGN neurites can navigate toward their target reliably regardless of where they are positioned within the three-dimensional structure of the ganglion.

Although it is well appreciated that astrocytes and microglia impact synapse formation[35], our results highlight an even earlier role for glia during axon outgrowth and guidance, before the glial precursors have differentiated. We find that in the cochlea, these glial precursors are not only along the migratory path, but also actively extending ahead of the SGN processes, forming chains that presage the appearance of radial bundles. Moreover, individual SGN processes grow differently when in contact with glial precursors, such that processes that are on glial precursors grow faster and make fewer changes in direction compared to those

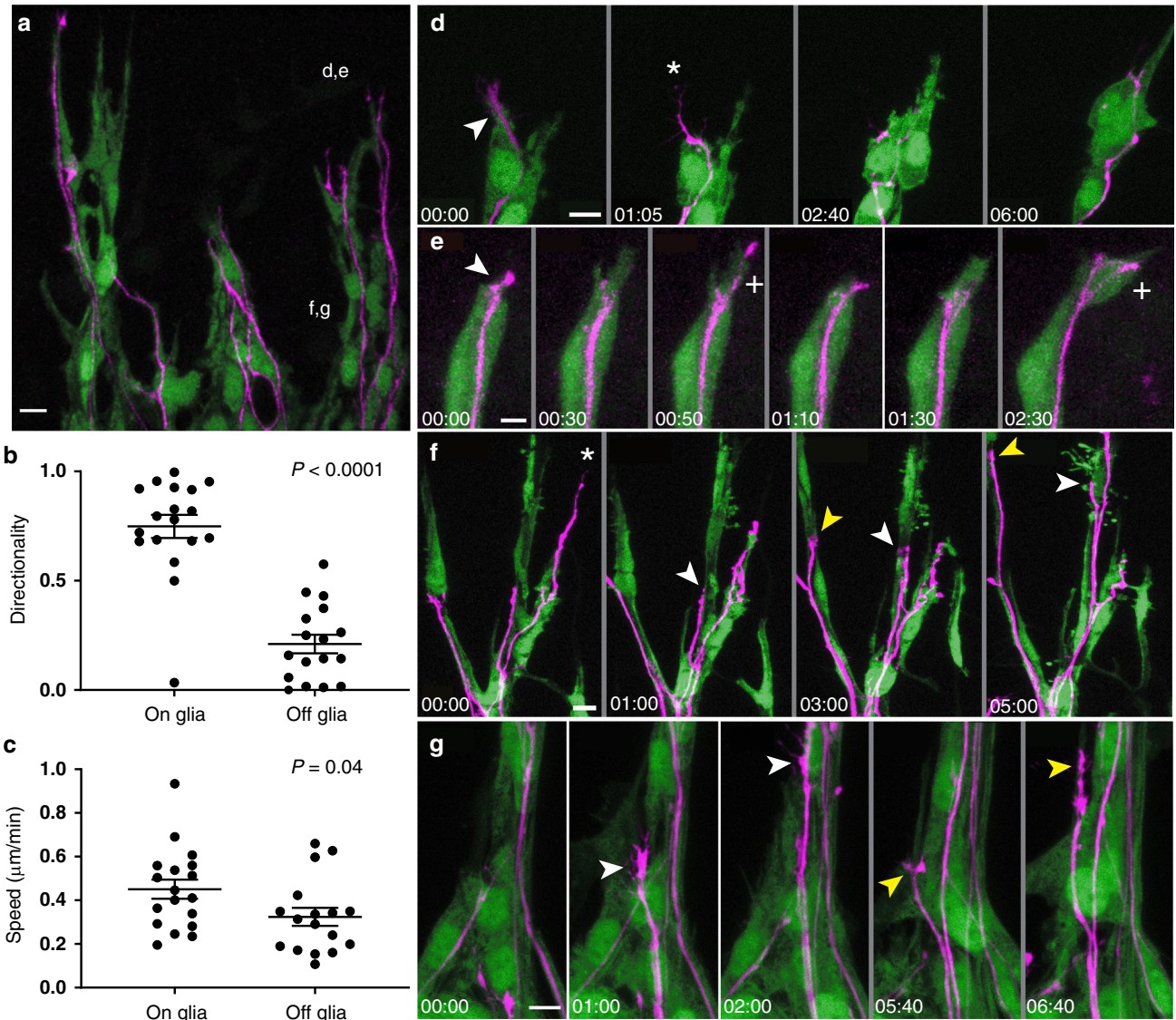

**Fig. 5 SGN processes alter their behavior when interacting with glial precursors. a** A low power view of the wavefront of peripheral process outgrowth in an E14 *Neurog1*$^{CreERT2}$*;AI14;PLP-GFP* cochlea, with SGN peripheral processes in magenta and glial precursors in green. SGN processes grow within chains of migrating glial precursors. The regions shown in (**d–g**) are indicated. **b, c** Directionality (**b**) and speed (**c**) for neurites growing on or off a glial precursor (raw data plotted, with mean ± SEM indicated). Process growth was more directed and faster on glial precursors. 18 on neurites and 17 off neurites (from 4 cochleae) were analyzed. Significance was assessed using an unpaired, two-tailed *t*-test. Directionality: on glia vs off glia, *P* < 0.0001. Speed: on glia vs off glia, *P* = 0.04. Source data are provided as a Source data file. **d, e** Frames from two movies at the wavefront, with time shown in hh:mm. In one movie (**d**, Supp. Movie 7), the distal tip of the SGN process (arrowhead, 00:00) is initially on top of a process from a glial precursor. By 01:05, the glial cell process has retracted (*). Subsequently, the SGN process also retracts (2:40) and then re-aligns along the second, remaining glial process (06:00). In the second movie (**e**, Supp. Movie 9), an SGN process starts on top of a glial precursor (arrowhead, 00:00). After the glial precursor extends forward (00:30), a branch from the SGN process follows (+, 00:50). The glial process retracts quickly (01:10) and then again extends forward (01:30), followed by the SGN neurite (+, 2:30). See Supplementary Movie 8 for an additional example of behavior at the wavefront. **f** A lower power movie from behind the wavefront (Supp. Movie 10) illustrates a retraction (*) and two examples of fasciculation along glial tracts (white and yellow arrowheads). **g** Frames from a movie behind the wavefront (Supp. Movie 11). Within a chain of glial precursors, one SGN peripheral process capped by a large growth cone moves straightforward in a short period of time (white arrowhead, 01:00 to 02:00). Subsequently, a second process (yellow arrowhead, 05:40 to 06:40) makes similarly rapid and directed process, possibly fasciculating along a thin PLP-GFP+ process ahead of it. Scale bars = 10 μm.

that are off. Thus, the SGN processes that are closest to the glial precursors, due to their position in the ganglion, can serve as pioneers for those that arrive later and thus encounter a different cellular environment. Importantly, it is clear that the SGN neurites do not require neural-crest-derived glial precursors to grow, given that neurite–neurite fasciculation also occurs and that SGN peripheral processes still eventually reach the organ of Corti in cochleas depleted of Schwann cells and satellite glia[13,28]. Our

data, together with published phenotypes, suggest instead that SGN neurites grow more efficiently in the presence of glial precursors.

The nature of neuron–glia interactions remains unclear. The glial precursors could work by actively promoting growth, for example by secretion of a relevant cue. Alternatively, they may stabilize filopodia extending from the SGN growth cone, for example by expression of an adhesion molecule. Although the

properties of glial precursors in the embryonic cochlea remain to be defined, the PLP-GFP+ cells that we have studied are likely precursors for Schwann cells, given their position, their similarities to glial precursors known to arise from the neural crest[26], and the fact that SGN neurites are disorganized when Schwann cells are absent but supporting cells in the organ of Corti remain[13]. However, it is also possible that there is some heterogeneity within this population, as suggested by the presence of particularly intense PLP-GFP+ cells surrounding the ganglion and in the funnel. Single cell RNA-sequencing studies may provide useful insights in the future.

Our observations support the classic idea that glial-guided growth of pioneer axons may be a general mechanism for establishing scaffolds throughout the nervous system. In fact, forty years ago, Singer et al. put forward the idea of a glial "blueprint" of the nervous system based on the close proximity of glial precursors and growth cones in the Newt spinal cord[36]. Likewise, in the cortex, glial processes form a "sling" straddling the lateral ventricles[37]. Similar to what we observed by imaging processes as they grow in the cochlea, glial processes appear to precede pioneer axons in the cortex as they cross the midline, with follower axons fasciculating. Additional evidence for glial-guided growth has come from genetic ablation of glia in flies, followed by analysis of longitudinal axons[38]. As in the cochlea, pioneer axons extend toward glia. When glia are absent, the growth cones move more slowly and some axons show altered fasciculation. A role in fasciculation has also been shown in studies of the developing lateral line in zebrafish[39], although in this case the glia seem to migrate along the axons.

Although the coincidence of glia and early axons is suggestive, it has been difficult to figure out exactly how the glia contribute to axon growth. When the glial sling is lesioned, the corpus callosum fails to form[37]. However, this does not mean that the glia instructed formation of the corpus callosum, as it is equally plausible that they play a permissive role. One challenge is the fact that glia also provide trophic support for neurons, so complete depletion of the glia can cause massive changes in neuron number and organization, making it hard to know whether early axon guidance phenotypes are primary or secondary[40]. Local ablation of glia, for instance in developing chicken embryos, offers one potential solution and has revealed minor axon guidance defects[26]. However, in these experiments, surrounding glia can proliferate and rescue the ablated area, again complicating interpretations. More definitive evidence that glia guide early axons has come from C. elegans, where glia are not required for neuronal survival[41]. Here, specific molecules have been identified and shown to act redundantly, underscoring the presence of multiple mechanisms to ensure accurate guidance, both of the pioneer axons along glia and of follower axons along the pioneers. An important next step will be to find the molecules that mediate similar interactions in the cochlea.

The use of glia to organize early axon outgrowth may complement canonical guidance systems and thereby improve the fidelity of circuit assembly. There is ample evidence that isolated axons can respond to instructive cues, so it is unlikely that glia are required for all guidance decisions. Instead, glia may collaborate with guidance systems, perhaps both permissively and instructively. One possibility is that the glial precursors respond to the same cues as the SGNs and simply provide a substrate that is more permissive for axon outgrowth. In this model, axon guidance errors are reduced by taking advantage of tiered levels of fasciculation: the early glial precursors pave the initial path for the leader axons, which in turn provide a surface for follower axons. This way, the later axons are more likely to find the right path, even as distances and cellular complexity increase. Unfortunately, testing such a model is not straightforward as one would not

expect axons to become completely lost but instead to be slightly slower or make more mistakes en route. Identifying these types of subtle guidance defects will require analysis of individual axon trajectories and more careful quantification, similar to how a role for floor plate-derived Netrin-1 was demonstrated[8,9]. It is also possible that the glial precursors themselves provide additional synergizing cues that directly impact neurite outgrowth behavior. Other types of glia do in fact express classic guidance cues, such as Semaphorins[42], and it will be interesting to test whether glial-derived guidance cues play any role during cochlear wiring in the future. Conversely, it will be important to learn how the glial precursors themselves are guided toward their targets. One possibility is that some of the glial precursors grow along SGN neurites. For instance, during the very earliest stages of outgrowth, before the spiral and vestibular ganglia have separated, neural processes seem to be just ahead of neural-crest-derived glial precursors[24]. It is possible that glia-guided growth of neurites occurs together with neurite-guided migration of glial precursors, such that reciprocal interactions between these populations further improve wiring reliability.

These findings have implications for efforts to re-wire the nervous system after damage. It has long been known that peripheral glia permit axon regeneration, whereas central glia are inhibitory[43]. Our work suggests that peripheral glia are not only permissive, but might even actively encourage axon re-extension. In animal models, SGN processes lose their synapses and retract following exposure to excessively loud sounds[44]. Similar loss of synapses has been observed in aged human cochleae, suggesting that damage to SGN processes accumulates over a lifetime of noise exposure and hence contributes to age-related hearing loss[45,46]. However, SGN peripheral processes persist despite being disconnected from the hair cells. One possibility is that the Schwann cells that remain could be used to stimulate re-extension to the hair cells. Importantly, Schwann cells are known to be unusually plastic and can de-differentiate to produce new cells in other regions of the nervous system[47]. Thus, it is possible that the cochlea could be re-wired by finding ways to de-differentiate the Schwann cells so that they regain their youthful outgrowth abilities.

## Methods

**Animals.** Bulk labeling of SGNs was accomplished by crossing *Bhlhb5^Cre* mice[48] to a Cre-dependent tdTomato reporter strain (*Ai14*; #007908 Jackson Laboratory)[49]. SGNs were sparsely labeled by crossing *Neurog1^CreERT2* mice[32] to the same reporter strain; this labeling reflects leaky Cre activity that occurs without administration of tamoxifen. Mice harboring a GFP reporter for PLP promoter activity (*PLP-GFP*) were used to label cochlear glia[34]. Animals were PCR genotyped using primers for Cre, tdTomato, or GFP; embryos were genotyped using a fluorescent dissecting microscope. To obtain timed pregnancies, male mice heterozygous for *PLP-GFP*, *Ai14*, and an appropriate *Cre* strain were bred with adult CD1 females (Charles River Laboratories). Noon on the day of the plug was considered embryonic day 0 (E0). Embryos and pups of either sex were used. Mice were maintained on a 12 h/12 h light/dark cycle at 18–23 °C and 40–60% humidity. Animals were maintained and handled ethically according to protocols approved by the Institutional Animal Care and Use Committee at Harvard Medical School.

**Organ culture and live imaging.** To collect and culture cochlear explants, fetuses were isolated from timed pregnant females, checked for fluorescence, and then dissected. Cochleae were placed into an imaging chamber, centered on a 1 mm hole punched into a small square of electrostatically charged cellulose membrane and held in place with a vitalline tissue drape. The imaging chamber contained culture media (defined DF12, Glutamax (Gibco), N2 supplement, 25 mM Glucose, 20 mM Hepes, 25 mM sodium bicarbonate, and 5% FBS (Gibco); no antibiotics are used) that was closed to atmosphere and heated to maintain the reported internal temperature at 37 °C. Cochleae were imaged along the mid-base region for each experiment. Using an Olympus FluoView 1000 confocal microscope, images were acquired every 7–14 min with a 60x oil lens through the entire area focal depth of interest (50–100 μm) in z-steps of 2–5 μm. Time-lapse files were tracked and analyzed with MetaMorph (Molecular Devices) and ImageJ software (NIH). Each time-lapse file was converted into RGB-depth coded movies to visualize individual neurite projections along the z axis. SGN processes were manually tracked and defined as any and all protrusions peripheral from the ganglion that

could be delineated from its neighbors and could be tracked for more than four time intervals. DiPER code[50] was used to analyze track speed and directionality. The XY coordinates for all tracks were recorded and plotted to show their orientation and start/stop points using Prism 8.1 (Graphpad).

Regions of interest were defined by measuring the position of the neurites relative to the border of the ganglion, delineated by the SGN cell bodies positioned there. Neurites in R1 were from ~E14.25-.5 cochleae and <50 μm from the border. R2 neurites were imaged in ~E15 cochleae and were 50–100 μm from the ganglion. R3 neurites were imaged in ~E15.5 cochleae and were >100 μm from the ganglion. Within the R1 and R2 regions, "wavefront" neurites were defined as those most distally extended at the time of imaging. Any neurites >10 μm modiolar to these distalmost neurites were classified as "behind."

The behavior of SGN neurites on and off glia was quantified by tracking neurites as described above while simultaneously monitoring the position of PLP-GFP labeled glia in a separate channel. "On" SGN neurites occupied a position that could not be resolved from the PLP-GFP signal in the glia. "Off" SGN neurites, by contrast, extended >1 μm away from PLP-GFP + glia for >2 time points.

**Immunostaining**. The following primary antibodies and dilutions were used in this study: Rabbit anti-DsRed (Clontech, 1:1000), Rabbit anti-TuJ (Biolegend, 1:1000), Goat anti-GFP FITC (Abcam 1:500), Chicken anti-Neurofilament (Millipore, 1:1000) and Chicken anti-GFP (Aves, 1:2000). Alexa-conjugated secondary antibodies raised in Donkey were all used at the same concentration (Thermo Fisher (anti-Goat A488) or Jackson ImmunoResearch (anti-Rabbit A488 and A594 and anti-Chicken A488; 1:500). For immunostaining, tissues were dissected, washed twice with PBS, and fixed with 4% paraformaldehyde (PFA) in phosphate buffered saline (PBS) overnight at 4 °C, and then washed three times in PBS.

For wholemount immunostaining, dissected cochleae were permeabilized and blocked at room temperature for 2 h in PBS containing 0.3% Triton, 3% bovine serum albumen, and 0.01% sodium azide. Primary antibody hybridization was performed in the same blocking buffer solution overnight at 4 °C, followed by three washes in PBS at room temperature. Secondary antibody hybridization was performed in blocking buffer overnight at 4 °C and extensively washed in PBS at room temperature the following day. Prior to imaging, tissues were cleared in BABB (benzyl alcohol:benzyl benzoate, 1:2 ratio). Briefly, cochleae were transferred through a graded series into 100% methanol and then placed flush on a glass slide within a rectangular silicon grease reservoir. Methanol within the reservoir was gently aspirated and replaced with a 1:1 solution of methanol:BABB. After partial clearing, this 1:1 solution was replaced with 100% BABB, and the silicon grease reservoir was sealed with a glass coverslip.

For section immunostaining, fixed E14-E14.5 heads were cryoprotected by incubating in 10% and 20% sucrose in PBS for one night each and then in 30% sucrose in NEG-50 (Richard-Allan Scientific) overnight, all at 4 °C. Heads were embedded in NEG-50 and stored at −80 °C prior to sectioning at 20 μm thickness. After sectioning, slides were allowed to dry at −80 °C overnight. Antigen retrieval in 10 mM citrate buffer (pH 6.0) was done for 20 min before commencing with the staining protocol. Blocking and antibody hybridization steps were carried out in PBS containing 5% Normal Donkey Serum and 0.5% Triton. Slides were blocked at room temperature for 1 h, followed by primary antibody hybridization overnight at 4 °C. Hybridization of the secondary antibody was carried out for 1 h at room temperature before washing thoroughly with PBS, incubating in DAPI and mounting a coverslip.

**Fixed tissue analysis**. After immunostaining and clearing, wholemount *Neurog1*CreERT2*;Ai14* cochleae (N = 7) were imaged on an Olympus FluoView 1000 confocal microscope. The entire volume of the mid-basal region of the ganglion was imaged in consecutive z-slices separated by 0.5 μm. These volumes were then rendered using Imaris. Neurons that could be clearly differentiated from adjacent cells and that could be viewed all the way from the cell body to the distalmost neurites were reconstructed using a combination of manual and automated methods that defined the outline of the cell plane by plane. The defined regions of each plane were then combined to create three-dimensional rendered surfaces. For analysis, the surfaces representing the cells of each region were subdivided by measuring the distance from the border to the rear of the cell group and dividing into three equal parts. Border (n = 28), mid (n = 52), and rear cells (n = 71) were assigned to groups based on the position of the center of the cell body. The primary process of each cell was considered to be the one which extended the greatest length from the cell body toward the organ of Corti. Imaris software was used to analyze each rendered cell to determine neurite length, defined as the total distanced covered; distance, defined as the linear distance between the neurite's origin and endpoint; directionality, defined as distance/length; z-displacement, defined as the distance a process traveled along the z axis between stacks; two slope values, calculated as z-displacement either over length or over distance; branch points, defined as the number of places where a branch of at least 5 μm splits off; and distance to branch, which is defined as the length from the process's endpoint to the nearest branch of at least 5 μm. Measurements were taken using a series of fixed points along the surfaces. The mean and SEM for each parameter were found for each of the three categories so that the characteristics of cell processes could be compared between regions. Only measurements for length, directionality, and slope are presented. Data from each of the seven cochleae are shown independently in Supplementary Fig. 2.

For analysis of neuron–glia interactions, wholemount *Neurog1*CreERT2*;Ai14; PLP-GFP* cochleae that had been stained for GFP, tdTomato and/or TuJ were cleared and then imaged on an Olympus FluoView 1000 confocal microscope, followed by analysis using Imaris. TdTomato+ or TuJ+ neurites in R1 were identified and then assessed for proximity to a PLP-GFP + glial cell, simply noting whether the neurite was ahead of the glial cell or behind it.

**Statistics**. Sample sizes were determined without any expectation of the effect size, but with cochleae from at least three different animals. All cells that could be confidently reconstructed or scored were included. One cochlea was excluded from the analysis in Fig. 2 due to incomplete labeling that prevented an unbiased assessment of morphology. SGN reconstructions were analyzed by an independent investigator blind to the possible outcomes. All statistical analyses were done using Prism 8.1 software (Graphpad). Data shown in Figs. 1, 4, and 5 were analyzed using an unpaired two-tailed *t*-test. Data in Fig. 2 were first assessed for normality and then analyzed using ANOVA with Tukey's multiple comparison test. Results with a $P < 0.05$ were considered statistically significant. Sample sizes and $P$ values are reported in the figure legends and are summarized in Supplementary Table 1.

**Reporting summary**. Further information on research design is available in the Nature Research Reporting Summary linked to this article.

## Data availability

Projections of the original reconstructed cochleae, examples of movies used to collect data, and the source data for graphs in Figs. 1–5 are provided with this paper. The relevant statistical analyses are summarized in Supplementary Table 1. For all quantifications, the raw data are shown along with mean and SEM. The original images and movies used to generate these data are available from the corresponding author upon request. Source data are provided with this paper.

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

## Acknowledgements

We thank D. Corey (Department of Neurobiology, Harvard Medical School) for providing access to his confocal microscope. We are also grateful to members of the Goodrich laboratory and Dr. Maxwell Heiman for helpful conversations and comments on the manuscript. This work was supported by the National Institutes of Health [R01 DC009223 to L.V.G.].

## Author contributions

This study was conceived of and designed by N.R.D. and L.V.G. N.R.D., E.B.H., O.O.O., and W.E.S. performed experiments and analyzed results. L.V.G. analyzed results and wrote the manuscript.

## Competing interests

The authors declare no competing interests.
