## [Peer Review File · Nature Communications]

Reviewers' Comments:

Reviewer #1:

Remarks to the Author:

Expertise: Glial and Neuronal interactions during development

Druckenbrod et al. present an interesting and exciting story dissecting the neuronal outgrowth of the cochlea. The study clearly identifies and fills an important gap in the axon guidance field by asking how glial cells contribute to the growth of axons. Their experimental model is perfectly designed to address the question as they can provide timelapse imaging movies of both glia and neurons as they are navigating. This study is a foundational study that will be cited for the basic glial and neuronal interactions that occur during outgrowth. It is an important study for later papers that dissect the molecular determinants of such glial and neuronal interactions. The impact of the paper, however, could be much higher if manipulations that test the observations were performed. Please see the suggested comments below to improve the manuscript:

1. The study demonstrates beautiful movies to support their claims. At times, however, these claims are overstated given the limitation of the experiment. For example, in Figure 6e, the neurons appear to extend short processes beyond the glia even though this is not noted in the paper. Is it possible that the glial front edge that is described is a consequence of the temporal and spatial resolution of the imaging? It could be argued that the neurons are extending fast transient "feelers" beyond the glia that simply are not captured with the resolution that was imaged. Is it also possible that neuronal extension precedes first but must be stabilized by glia behind it, and the imaging is capturing the stabilization component This is important, as it changes the model of how outgrowth could occur.

2. The study confirms many important aspects of glial interactions during neuronal outgrowth but the potential impact of the study could be so much higher. To strengthen the claims in the paper and increase the impact, a manipulation of either the glial or neurons could be performed. For example, does laser ablating the glia cause the neuron to misguide. Or does it slow neuronal outgrowth? A manipulation is critical as it strengthens the hypotheses in the paper that are currently supported by time-lapse imaging of wildtype animals. I recognize these are challenging experiments but nonetheless important. Such experiments would also help to address the above comment of whether the glia are stabilizing the neurons or functioning as guideposts.

Minor Specific points:

Can the authors clarify that the glia they are looking at are Schwann cells vs Satellite glia based on the PLP reporter?

Figure 1 is not cited at the beginning of the results.

In figure 2, the authors note that neuronal processes and glia interact, at least during the earliest stages. Did this extend beyond citation 23 and 24.

The authors note "Time-lapse imaging in embryonic cochlear explants confirmed..." but do not reference a figure or citation. Was this work completed by the authors or others?

I commend the great analysis showing how the axons grow toward their targets. This information is integral in models of inner ear development.

In figure 2d, it appears that some neurites extend past the glial wavefront, if even for short distances. Perhaps it is more accurate to state that glial cells are present ahead of a subset of neuronal processes

Reviewer #2:

Remarks to the Author:

The manuscript by Druckenbrod et al. is a beautifully written summary of the results of immensely challenging imaging studies of neuron-glia interactions during early phases of cochlear innervation. The carefully curated time-lapse imaging data provides unequivocal evidence of glial-guided neuronal growth. Glia had not previously been shown to be major players in this process, so the data are novel, and I expect this study to be well-cited by the auditory and neuroscience fields. In addition, since glia are easy to grow and are likely easy to transplant, glia could represent a useful growth substratum for neuronal regrowth in a damaged cochlea. Given the difficulty of these imaging experiments, I'm impressed at the depth and thoroughness of sampling and analysis. I have no significant concerns about this manuscript prior to publication – only a few minor comments below.

1. Throughout the manuscript, there are morphometric parameters that could use more clearly-stated definitions in the narrative of the Results sections. Examples are "tortuosity" and "directionality."
2. In the second paragraph of the Results section, the statement that the neurons "lie on top" seems vague. The authors should also include a few more details on the anatomical position of these fibers as they would appear in the intact embryo.
3. The x-axis of figure 2G is uninformative. I think the authors mean "% of neurites behind glia."
4. Figure 3b. It appears that some neurons are oriented along the longitudinal axis of the cochlea. Are they migrating? The authors should note whether any of the neurons they imaged appeared to be migrating along the length of the cochlea.
5. In the 3rd paragraph of the introduction, the authors reference Semaphorin-3A, when the manuscript they cited in this context focuses on Semaphorin-3F. Please clarify.

Reviewer #3:

Remarks to the Author:

This study from the laboratory of Lisa Goodrich at Harvard seeks to examine the interactions between the extension of spiral ganglion processes and glial cells during inner ear development. The images are striking and the analysis is consistent with the high standards of the Goodrich lab. However, the results are entirely correlative or, in some cases possibly phenomenological. As a result, the main conclusion of the study, that glia play a guiding role in directing process outgrowth is not supported by the data. Are the two cell types closely associated during early process outgrowth, for sure, but can one conclude that the glia are taking the lead, I don't think so. It is just as likely that the glia are following along behind the pioneering processes. The demonstration of different behaviors for different processes based on their distance from the edge of the ganglion is quite nice and perhaps provides a glimpse into changes in process extension and behavior with age, assuming neurons located farthest from the border are the oldest.

Specific comments:

Page 6, end of second paragraph: the notion of neurons lying on top of glia is hard to conceive in a three dimensional structure. In fact in Figure 2c and c' it would seem that there is an intermingling between the neurons and the glia. Wouldn't it be more accurate to say that the processes grow within the glia rather than "on top"?

Also, relevant to the rest of the study, in this section there is clearly at least one process that has extended past the edge of the glia.

Fig. 2g: While these data support the idea that the vast majority of processes are growing on glia, it is important to consider that the sparse labeling only shows a subset of the neurons. Therefore, 20% of the processes in the images, most of which are unlabeled, have extended past the glia. In this case, couldn't both the remaining processes and the glia be following those first, pioneer, processes?

Page 8, end of first paragraph, similar to above, the use of the phrase "glia carpet" seems inaccurate. Isn't it more of a "glial tube"?

Also on page 8. The first paragraph discusses the behavior of processes as they extend within the border of the ganglion. But then the first sentence in the next paragraph states "Time lapse imaging in embryonic cochlear explants confirmed that SGN peripheral processes change their outgrowth behavior as they progress from the ganglion toward the developing organ of Corti". The inclusion of the word "confirmed" implies that this is a similar comparison to the previous paragraph, but isn't one analysis looking at processes within the borders of the ganglion while the time lapse imaging is looking at behavior outside the ganglion?

Page 10, heading includes "early born" in the title. Is this relevant? If so, how is this known? I don't disagree that these are most likely the earliest born neurons, which explains why their processes are the longest and straightest, just wondering how this is known?

Same paragraph, is it possible that the differences in trajectory behavior are a result of the age of the neuron rather than its position?

Response to Referees

We thank the reviewers for their helpful and insightful comments. In response to the points raised, we have re-written the text significantly and also revised Figures 2 and 3 as requested. We feel that the revised manuscript communicates our results and interpretations more fairly and clearly with these improvements. Although we had hoped to provide new data to further support our model, the most definitive experiments are not possible within a reasonable time frame. We tried to further test the model using less precise methods, but unfortunately, these experiments came to a halt due to the current pandemic and shutdown of research. We hope that the reviewers agree that even without these data, the revised manuscript presents a compelling and rigorous analysis of neuron-glia interactions and their potential role in cochlear wiring. Specific responses to each point are provided below. Changes are indicated in blue in the revised text. Note also that we updated Supplemental Table 1 to include all information requested regarding the statistical analysis.

Reviewer #1 (Remarks to the Author):

Druckenbrod et al. present an interesting and exciting story dissecting the neuronal outgrowth of the cochlea. The study clearly identifies and fills an important gap in the axon guidance field by asking how glial cells contribute to the growth of axons. Their experimental model is perfectly designed to address the question as they can provide timelapse imaging movies of both glia and neurons as they are navigating. This study is a foundational study that will be cited for the basic glial and neuronal interactions that occur during outgrowth. It is an important study for later papers that dissect the molecular determinants of such glial and neuronal interactions.

Thank you for the enthusiastic support of this study.

The impact of the paper, however, could be much higher if manipulations that test the observations were performed. Please see the suggested comments below to improve the manuscript:

1. The study demonstrates beautiful movies to support their claims. At times, however, these claims are overstated given the limitation of the experiment. For example, in Figure 6e, the neurons appear to extend short processes beyond the glia even though this is not noted in the paper. Is it possible that the glial front edge that is described is a consequence of the temporal and spatial resolution of the imaging? It could be argued that the neurons are extending fast transient "feelers" beyond the glia that simply are not captured with the resolution that was imaged. Is it also possible that neuronal extension precedes first but must be stabilized by glia behind it, and the imaging is capturing the stabilization component This is important, as it changes the model of how outgrowth could occur.

The reviewer makes the important point that one must always wonder what is not seen when using these kinds of techniques. We attempted to alleviate this concern in the original submission with the data in Figure 2, which are from fixed specimens. These data show a high proportion of neurites, stained using two different methods, behind the glia. Thus, we can at least be confident

Response to Referees

that most of the main neurites are behind the glia. We acknowledge that we cannot rule out the possibility that extremely fine processes are present that are not filled either with tdTomato or by anti-TuJ1 staining. However, short of electron microscopy, there is no straightforward way to know for sure that such processes do not exist. We added the following statement to the Results to make this point more clearly:

“Since similar percentages were observed when quantifying all β -III tubulin+ processes or only those randomly labeled by Neurog1^{CreERT2} recombination of *Ai14*, the behavior of tdTomato+ processes seems to be typical of all SGN processes. Additionally, since both methods yielded similar results, it is likely that we are visualizing the entire SGN terminal, though we cannot rule out the presence of very fine processes that are not filled with either tdTomato or β -III tubulin.” (p. 7)

That said, the behavior of the main neurite is highly relevant. As shown in Figure 6, these neurites are more directed and grow faster when contacting glia. Regardless of what might be happening ahead of the process, these observations provide strong evidence that interactions with glia affect neurite behavior in ways that match what we see in fixed tissue. Finally, as noted by the reviewer, neurites absolutely do extend beyond and/or off glia, as is emphasized by the quantifications in Fig. 6b and 6c. We did not mean to suggest that neurites cannot grow without glia. This is clearly not the case, given the fact that SGN peripheral processes are still able to reach the organ of Corti in *ErbB2* mutants. Instead, the model is that this growth is facilitated by glia. We also did not mean to infer anything about the mechanism of action, i.e. whether the glia are reaching out and pulling the SGN neurites towards them versus simply stabilizing any filopodia that come in contact with them. We have revised the text to emphasize that we do not think that glia are necessary for growth *per se* and to present multiple possibilities for how glia might affect neurite growth:

“Importantly, it is clear that the SGN neurites do not require glia to grow, given that neurite-neurite fasciculation also occurs and that SGN peripheral processes still eventually reach the organ of Corti in cochleas with no glia. Our data, together with published phenotypes, suggest instead that SGN neurites grow more efficiently in the presence of glia. The nature of these interactions remains unclear. For example, the glia could work either by actively promoting growth, for example by secretion of a relevant cue, or by stabilizing filopodia extending from the SGN growth cone, for example by expression of an adhesion molecule.” (p. 14)

2. The study confirms many important aspects of glial interactions during neuronal outgrowth but the potential impact of the study could be so much higher. To strengthen the claims in the paper and increase the impact, a manipulation of either the glial or neurons could be performed. For example, does laser ablating the glia cause the neuron to misguide. Or does it slow neuronal

Response to Referees

outgrowth? A manipulation is critical as it strengthens the hypotheses in the paper that are currently supported by time-lapse imaging of wildtype animals. I recognize these are challenging experiments but nonetheless important. Such experiments would also help to address the above comment of whether the glia are stabilizing the neurons or functioning as guideposts.

We agree wholeheartedly. As noted in the manuscript, there is already published evidence that the pattern of SGN innervation is disrupted in mice with depleted glia. However, loss of glia is not a clean enough manipulation, as the absence of cell bodies as well as the trophic factors they provide could affect neurite growth secondarily. Additionally, we cannot rule out effects of the SGN cell body position, which is also abnormal in *ErbB2* and *Sox10* mutants. To work around these problems, we considered laser ablation, as proposed, but this is tricky, as there are so many neurites and glia packed together that it would be necessary to ablate a whole swath of glia to be able to generate interpretable results. Additionally, the presence and then clearance of the debris would further complicate interpretations. A better way to test our model is to maintain the same number of glia but to freeze them in place in the mesenchyme. We spent the past several months trying to develop a method to arrest glial growth specifically in cultured embryonic cochlear explants. Unfortunately, these are tricky experiments and our progress was halted by the shutdown of research at Harvard. At this point, it will take months to get these experiments to work at all and more time after that to analyze the results.

In the meantime, we considered testing causality by conditional ablation of the glia or of ErbB3 signaling at specific timepoints, working with Dr. Gabriel Corfas, who has the relevant mouse lines. Such a manipulation might at least avoid complications from mispositioned SGN soma, if the timing is just right. Again, we were unable to complete a pilot study due to the research shutdown. Although such studies could provide some hints, any of these manipulations would only modestly advance what is already known from our study and from previous analysis of *ErbB2* and *Sox10* mutants. In fact, the *ErbB2* and *Sox10* phenotypes indicate that growth itself can still occur, just maybe not as reliably or efficiently. Thus, we would need to perform several years' worth of additional imaging experiments in order to reach firm conclusions about any impact on SGN neurite growth behavior. Not only are the imaging experiments and analyses themselves time-intensive, but it would also be necessary to cross both *Neurog1^{CreERT2}* and *Ai14* into the mutant background, with very low odds for collecting controls and mutants with labeled SGNs. Further, in these mutants, the SGNs would still be abnormally positioned, further complicating interpretation.

With these various issues in mind, we feel it is more fruitful to define the molecular basis of neuron-glia interactions. With the identification of relevant molecules, we can design and carry out much more definitive experiments than what is currently possible. We are currently searching for relevant molecules, but it will take several years to complete this line of investigation. Given the inherent limitations of the types of studies needed to address their concerns, we hope that the reviewers agree that the current study is beneficial to *Nature*

Response to Referees

Communications readers as is. To avoid the impression of overstating the findings or suggesting that we have defined causality, we have revised the text throughout to make it clearer what is correlative and why it will be tricky to know for sure how these interactions work until we know what molecules are involved. See for example the paragraphs from p. 15-16 (“Although the coincidence of glia and early axons is suggestive, it has been difficult to figure out exactly how the glia contribute to axon growth...” and “The use of glia to organize early axon outgrowth...”).

Can the authors clarify that the glia they are looking at are Schwann cells vs Satellite glia based on the PLP reporter?

At E14.5, the glia are likely progenitors that could produce either Schwann cells or satellite glia, depending on where they end up. The text has been revised accordingly:

“At this stage, it is not yet possible to tell which of the PLP-GFP+ progenitors will differentiate as Schwann cells or satellite cells.” (p. 6)

Figure 1 is not cited at the beginning of the results.

We refer to Figure 1 in the Introduction, as it explains relevant features of the anatomy to readers who might not be familiar with cochlear wiring.

In figure 2, the authors note that neuronal processes and glia interact, at least during the earliest stages. Did this extend beyond citation 23 and 24.

To our knowledge, the papers by Carney and Silver (1983) and Sandell et al., (2014) are the only ones to report the anatomical relationship between glia and SGN processes when they are first present.

The authors note "Time-lapse imaging in embryonic cochlear explants confirmed..." but do not reference a figure or citation. Was this work completed by the authors or others?

We apologize for the lack of clarity. We were trying to make the point that time lapse imaging results confirmed what we inferred from analysis of fixed tissue, but in re-reading the sentence now, we can see why this was confusing. The sentence has been re-written:

“As predicted from analysis of fixed tissue, time lapse imaging in embryonic cochlear explants revealed that SGN peripheral processes change their outgrowth behavior as they progress from the ganglion toward the developing organ of Corti.” (p.8)

I commend the great analysis showing how the axons grow toward their targets. This information is integral in models of inner ear development.

Thank you.

Response to Referees

In figure 2d, it appears that some neurites extend past the glial wavefront, if even for short distances. Perhaps it is more accurate to state that glial cells are present ahead of a subset of neuronal processes

Yes, this is a much better way of stating what we observe. The sentence has been revised to read:

“Quantification showed that most peripheral processes are preceded by glia : $72.25\% \pm 5.63$ (s.e.m.) of the most distal tdTomato-labeled SGN peripheral in the E14 cochlea were preceded by a PLP-GFP positive glial cell (n=125 neurites from 6 cochleae) (Fig. 2g).” (p. 7)

Reviewer #2 (Remarks to the Author):

The manuscript by Druckenbrod et al. is a beautifully written summary of the results of immensely challenging imaging studies of neuron-glia interactions during early phases of cochlear innervation. The carefully curated time-lapse imaging data provides unequivocal evidence of glial-guided neuronal growth. Glia had not previously been shown to be major players in this process, so the data are novel, and I expect this study to be well-cited by the auditory and neuroscience fields. In addition, since glia are easy to grow and are likely easy to transplant, glia could represent a useful growth substratum for neuronal regrowth in a damaged cochlea. Given the difficulty of these imaging experiments, I'm impressed at the depth and thoroughness of sampling and analysis. I have no significant concerns about this manuscript prior to publication – only a few minor comments below.

Thank you very much for the kind words and recognition of how hard these experiments are.

1. Throughout the manuscript, there are morphometric parameters that could use more clearly-stated definitions in the narrative of the Results sections. Examples are “tortuosity” and “directionality.”

This is a helpful suggestion. In response, we added the following statement:

“Speed was calculated by dividing the distance between the origin and the final position over time. Directionality was quantified as the length of the entire path divided by the length from origin to final position.” (p. 9)

We also changed the x-axis title in Fig. 3c from “Tortuosity” to “Directionality”, which means the same thing. We did not see any other use of the word “tortuosity.”

2. In the second paragraph of the Results section, the statement that the neurons “lie on top” seems vague. The authors should also include a few more details on the anatomical position of these fibers as they would appear in the intact embryo.

Response to Referees

We have revised the text to better explain the position of the neurites. It is tricky to visualize, as in wholemound preparations, the neurites basically grow on top of the glia, which are closest to the basilar membrane. However, in three-dimensions, the neurites are more likely growing through a glial “funnel”, the term originally used by Carney and Silver. We have adopted this term and also tried to make the anatomical organization clearer throughout the paper.

3. The x-axis of figure 2G is uninformative. I think the authors mean “% of neurites behind glia.”

We apologize for the confusion. The x-axis has been revised as suggested.

4. Figure 3b. It appears that some neurons are oriented along the longitudinal axis of the cochlea. Are they migrating? The authors should note whether any of the neurons they imaged appeared to be migrating along the length of the cochlea.

This is an interesting question. At E14.5, the cochlea is still extending, so some of the SGNs may appear longitudinal as a result; they may also still be polarizing. During imaging at these stages, we occasionally observed medial to lateral displacement of groups of SGN soma. This may be due to local tissue growth, as groups of surrounding cells were affected. Translocation of individual SGN somas was not observed. We have added this point to the Results:

“Because the cochlear duct is still lengthening at this stage, groups of SGNs occasionally moved, consistent with the appearance of a few longitudinal cell bodies in fixed tissue (Fig. 3b). However, we did not observe any translocation of individual SGN somas that would indicate active migration along the length of the cochlea.” (p. 10).

5. In the 3rd paragraph of the introduction, the authors reference Semaphorin-3A, when the manuscript they cited in this context focuses on Semaphorin-3F. Please clarify.

Our mistake – this was a typo that has now been corrected.

Reviewer #3 (Remarks to the Author):

This study from the laboratory of Lisa Goodrich at Harvard seeks to examine the interactions between the extension of spiral ganglion processes and glial cells during inner ear development. The images are striking and the analysis is consistent with the high standards of the Goodrich lab. However, the results are entirely correlative or, in some cases possibly phenomenological. As a result, the main conclusion of the study, that glia play a guiding role in directing process outgrowth is not supported by the data. Are the two cell types closely associated during early process outgrowth, for sure, but can one conclude that the glia are taking the lead, I don't think so. It is just as likely that the glia are following along behind the pioneering processes.

Response to Referees

As noted in our response to the first reviewer, we agree that the data are correlative and we tried to make this clear throughout the study. It is very hard to demonstrate causality, unfortunately, as highlighted by a paragraph in the Discussion (“Although the coincidence of glia and early axons is suggestive...”). From the work of Mao et al and Morris et al, we already know that neurites do not form radial bundles when there are no glia. Whether this is because the glia are actively eliciting outgrowth or because they provide some kind of stabilizing cue is impossible to say without many more experiments. We tried to make this clearer by adding the following text to the Discussion:

“Importantly, it is clear that the SGN neurites do not require glia to grow, given that neurite-neurite fasciculation also occurs and that SGN peripheral processes still eventually reach the organ of Corti in cochleas with no glia^{25,26}. Our data, together with published phenotypes, suggest instead that SGN neurites grow more efficiently in the presence of glia. The precise nature of these interactions remains unclear. For example, the glia could work either by actively promoting growth, for example by secretion of a relevant cue, or by stabilizing filopodia extending from the SGN growth cone, for example by expression of an adhesion molecule.” (p. 14)

As summarized above, we attempted to test causality using some *in vitro* assays and newly available glia mutants (in collaboration with the Corfas lab). Unfortunately, the research shutdown at Harvard has made it impossible to complete these studies for at least another year. Additionally, even if successful, many time-consuming time lapse imaging studies would be needed to conclude that the glia are taking the lead. As such, we feel that the current movies together with the analysis in fixed tissue provide strong evidence glia could actively guide SGN neurites, given their position (Fig 2) and the strikingly different behavior of neurites that are contacting or not contacting glia (Fig 7). We have revised the text to make it clear that additional experiments are needed to test the model definitively. For instance, we revised the last paragraph of the Introduction to set the readers’ expectations more appropriately:

“...we show that SGN neurites interact with each other and with glia during radial bundle formation. The timing and nature of these interactions suggest that neuron-glia interactions establish a scaffold for subsequent neuron-neuron interactions.”
(p. 5)

We also added a phrase in the Discussion noting that “it is unlikely that glia are required for all guidance decisions” (p. 16) and emphasized that “Unfortunately, testing such a model is not straightforward...” (p. 16). There is also a paragraph dedicated to the problems of demonstrating causality more broadly in the nervous system (p. 15).

Response to Referees

In terms of the possibility that the glia follow the pioneering processes, we did not mean to imply that reciprocal interactions do not occur. Indeed, it seems quite likely that some of the glia grow along SGN neurites (though at the stages we have analyzed, this seems rare). If there is some SGN-guided glial growth that sets everything off, this would not diminish the role that subsequent interactions play in encouraging fast and directed growth towards the organ of Corti, as indicated by the analysis in Fig. 7. We thank the reviewer for emphasizing this possibility, which is now indicated more explicitly in the revised discussion:

“One interesting possibility is that some of the glia are grow along SGN neurites. For instance, during the very earliest stages of outgrowth, before the spiral and vestibular ganglia have separated, neural processes seem to be just ahead of neural crest-derived glia. It is possible that glia-guided growth of neurites occurs together with neurite-guided migration of glia, such that reciprocal interactions between these populations further improve wiring reliability.” (p. 16-17)

The demonstration of different behaviors for different processes based on their distance from the edge of the ganglion is quite nice and perhaps provides a glimpse into changes in process extension and behavior with age, assuming neurons located farthest from the border are the oldest.

Thank you.

Specific comments:

Page 6, end of second paragraph: the notion of neurons lying on top of glia is hard to conceive in a three dimensional structure. In fact in Figure 2c and c' it would seem that there is an intermingling between the neurons and the glia. Wouldn't it be more accurate to say that the processes grow within the glia rather than "on top"?

The other reviewers made similar points and we have revised the language throughout the paper accordingly. For instance:

“...the SGN peripheral processes are closely associated with a distinct population of glia” (p. 6)

“...the earliest SGN peripheral processes extend along intensely PLP-GFP-positive glia.” (p. 6)

Also, relevant to the rest of the study, in this section there is clearly at least one process that has extended past the edge of the glia.

This point was also made by the other reviewers and we sincerely apologize for the lack of clarity. We did not mean to suggest that neurites never grow past the glia. If this were true, then the graphs in Fig. 2 would be at 100%. Our model is that when neurites grow on glia, they grow faster and more directed and that this help with radial bundle formation. These data suggest that glia-guided SGN neurite outgrowth can occur. We did not mean to suggest that this is an obligate

Response to Referees

order, where all SGN neurites must follow glia. Clearly, neurites can grow without glia, as argued above, since neurites grow on neurites and can also reach the organ of Corti even in mutants lacking glia, just in a disorganized fas. We have revised the text to make the model clearer:

“Qualitatively, the neurites and glia seemed to grow in tandem in a dynamic fashion, with neurites occasionally observed without any nearby glia and vice versa. This is consistent with what we observed in fixed tissue, where ~20% of neurites extended beyond glia at the wavefront (Fig. 2g).

To gain a better sense of how peripheral process growth might be influenced by interactions with the glia, we quantified the speed and directionality of neurites while they were growing either on or off glia, as defined by co-localization of tdTomato and GFP signals. Quantification revealed that the neurites grew faster and in a directed manner when in contact with glia (Fig. 6b,c). By contrast, SGN processes that were not contacting glia were slower and more exploratory, with a significantly lower direction index (Fig. 6b,c). Thus, SGN processes make a more efficient path towards the organ of Corti when they are interacting with glia.” (p. 12)

Fig. 2g: While these data support the idea that the vast majority of processes are growing on glia, it is important to consider that the sparse labeling only shows a subset of the neurons. Therefore, 20% of the processes in the images, most of which are unlabeled, have extended past the glia. In this case, couldn't both the remaining processes and the glia be following those first, pioneer, processes?

We had the same thought, which is why we also analyzed all anti-TuJ1 labeled processes, not just those labeled with tdTomato (see Fig. 2g). These data suggest that the behavior of the tdTomato+ neurites is reflective of the overall population, since the percentages are similar. We added a statement to the Results to make this clearer (see response to Reviewer 1). We also note that the fact that some neurites grow past the glia does not invalidate the model. Those neurites that grow past the glia may well be pioneers that grew independent of glia. Alternatively, they may be exploratory neurites that will retreat and then be gently redirected by glia, as is suggested by some of our time lapse imaging studies. We really don't have a way to distinguish among these possibilities without imaging from the very very earliest stages and keeping them growing for several days. This is not possible technically yet. We apologize for making it seem as if we feel that neurite growth is entirely dependent on glia and have revised the text to make the model clearer:

“One interesting possibility is that some of the glia are grow along SGN neurites. For instance, during the very earliest stages of outgrowth, before the spiral and vestibular ganglia have separated, neural processes seem to be just ahead of neural

Response to Referees

crest-derived glia. It is possible that glia-guided growth of neurites occurs together with neurite-guided migration of glia, such that reciprocal interactions between these populations further improve wiring reliability.” (p. 16-17)

Page 8, end of first paragraph, similar to above, the use of the phrase “glia carpet” seems inaccurate. Isn’t it more of a “glial tube”?

This is a fair point and one raised by the other reviewers as well. We used the term carpet because of their appearance in 3D reconstructions, but other researchers have used the term funnel, which was coined by Carney and Silver is perhaps more accurate. We have adopted this term throughout the text accordingly.

Also on page 8. The first paragraph discusses the behavior of processes as they extend within the border of the ganglion. But then the first sentence in the next paragraph states “Time lapse imaging in embryonic cochlear explants confirmed that SGN peripheral processes change their outgrowth behavior as they progress from the ganglion toward the developing organ of Corti”. The inclusion of the word “confirmed” implies that this is a similar comparison to the previous paragraph, but isn’t one analysis looking at processes within the borders of the ganglion while the time lapse imaging is looking at behavior outside the ganglion?

We apologize for the misleading wording, which was also noted by Reviewer 1. We revised the sentence as follows:

“As predicted from analysis of fixed tissue, time lapse imaging in embryonic cochlear explants revealed that SGN peripheral processes change their outgrowth behavior as they progress from the ganglion toward the developing organ of Corti.” (p. 8)

Page 10, heading includes “early born” in the title. Is this relevant? If so, how is this known? I don’t disagree that these are most likely the earliest born neurons, which explains why their processes are the longest and straightest, just wondering how this is known?

Same paragraph, is it possible that the differences in trajectory behavior are a result of the age of the neuron rather than its position?

Thank you for pointing this out. This is an accidental holdover from an earlier version of the manuscript. It is certainly possible that differences in trajectory behavior relate to age rather than position, but we do not have any data that would allow us to comment on anything but position at this time.

Reviewers' Comments:

Reviewer #1:

Remarks to the Author:

The authors have done an excellent job of addressing the reviews. I commend them for their articulate and thoughtful response to the critiques of the initial submission. The additional text that was added to the manuscript nicely corroborates the data and conclusions that are presented. I particularly commend them for attempting to experimentally manipulate the system to test their hypotheses. While this was ultimately unsuccessful, their text addresses the concerns I initially outlined in the review. I now support the publication of this work in Nature Comm.

Reviewer #2:

Remarks to the Author:

I am satisfied with the changes the authors have made in response to my minor comments from the first round of review. One thing though -- the addition of the commentary regarding SGNs migrating doesn't appear in the revised manuscript. The authors indicated that they added this paragraph:

"Because the cochlear duct is still lengthening at this stage, groups of SGNs occasionally moved, consistent with the appearance of a few longitudinal cell bodies in fixed tissue (Fig. 3b). However, we did not observe any translocation of individual SGN somas that would indicate active migration along the length of the cochlea." (p. 10).

I wasn't able to see it. Perhaps it was deleted to try and limit word count? The authors should try and squeeze it in somewhere because it is a good point.

Reviewer #3:

Remarks to the Author:

My primary concern with the original version of this manuscript was the correlative nature of the experiments. The authors agree with this limitation and fairly point out that direct experiments would be challenging to do, in particular given the current situation in terms of the ability to generate new data.

With that consideration, they have done a nice job of addressing and qualifying their conclusions based on the acknowledged limitations of the results.

Point by point response

From Reviewers:

Reviewer #1 (Remarks to the Author):

The authors have done an excellent job of addressing the reviews. I commend them for their articulate and thoughtful response to the critiques of the initial submission. The additional text that was added to the manuscript nicely corroborates the data and conclusions that are presented. I particularly commend them for attempting to experimentally manipulate the system to test their hypotheses. While this was ultimately unsuccessful, their text addresses the concerns I initially outlined in the review. I now support the publication of this work in Nature Comm.

We thank the reviewer for their understanding.

Reviewer #2 (Remarks to the Author):

I am satisfied with the changes the authors have made in response to my minor comments from the first round of review. One thing though -- the addition of the commentary regarding SGNs migrating doesn't appear in the revised manuscript. The authors indicated that they added this paragraph:

“Because the cochlear duct is still lengthening at this stage, groups of SGNs occasionally moved, consistent with the appearance of a few longitudinal cell bodies in fixed tissue (Fig. 3b). However, we did not observe any translocation of individual SGN somas that would indicate active migration along the length of the cochlea.” (p. 10).

I wasn't able to see it. Perhaps it was deleted to try and limit word count? The authors should try and squeeze it in somewhere because it is a good point.

Thank you for the support and for pointing out that the discussion of cell body movement was not included. This information has now been added (p. 9).

Reviewer #3 (Remarks to the Author):

My primary concern with the original version of this manuscript was the correlative nature of the experiments. The authors agree with this limitation and fairly point out that direct experiments would be challenging to do, in particular given the current situation in terms of the ability to generate new data.

With that consideration, they have done a nice job of addressing and qualifying their conclusions based on the acknowledged limitations of the results.

We thank the reviewer for their understanding.